# Evaluation of a novel, rapid antigen detection test for the diagnosis of SARS-CoV-2

**Rainer Thell**[1,2]*, **Verena Kallab**[1,2], **Wolfgang Weinhappel**[3], **Wolfgang Mueckstein**[3], **Lukas Heschl**[4], **Martina Heschl**[4], **Stefan Korsatko**[5], **Franz Toedling**[6], **Amelie Blaschke**[1,2], **Theresa Herzog**[1,2], **Anna Klicpera**[1,2], **Clara Koeller**[7,8], **Moritz Haugk**[1,7], **Anna Kreil**[1,9], **Alexander Spiel**[1,10], **Philipp Kreuzer**[11], **Robert Krause**[12], **Christian Sebesta**[1,2], **Stefan Winkler**[13], **Brenda Laky**[14,15], **Marton Szell**[1,2]*

1 Wiener Gesundheitsverbund, Vienna, Australia, 2 Department of Internal Medicine 2, Emergency Department, Klinik Donaustadt, Vienna, Austria, 3 Primary Health Care Centre Medizin Mariahilf, Vienna, Austria, 4 Primary Health Care Centre Landarztteam, Oed, Austria, 5 Primary Health Care Centre Medius, Graz, Austria, 6 Primary Health Care Centre Praxis Dr Toedling, Probstdorf, Austria, 7 Emergency Department, Klinik Hietzing, Vienna, Austria, 8 Semmelweis University Budapest, Hungary, 9 Emergency Department, Klinik Landstrasse, Vienna, Austria, 10 Emergency Department, Klinik Ottakring, Vienna, Austria, 11 Emergency Department, Univ. Clinic of Internal Medicine, Medical University Graz, Graz, Austria, 12 Univ. Clinic of Internal Medicine, Section of Infectiology and Tropical Medicine, Medical University Graz, Graz, Austria, 13 Department of Infectiology and Tropical Medicine, University Clinic of Internal Medicine I, Medical University Vienna, Austria, 14 MedSciCare, Vienna, Austria, 15 Competence Centre Clinical Research, University Clinic of Dentistry, Medical University of Vienna, Vienna, Austria

* rainer.thell@meduniwien.ac.at (RT); marton.szell@gesundheitsverbund.at (MS)

**Data Availability Statement:** All relevant data are within the paper and its Supporting information files.

## Abstract

### Background

Severe acute respiratory syndrome coronavirus 2 (SARS-CoV-2) causing coronavirus disease 2019 (COVID-19) is currently finally determined in laboratory settings by real-time reverse-transcription polymerase-chain-reaction (rt-PCR). However, simple testing with immediately available results are crucial to gain control over COVID-19. The aim was to evaluate such a point-of-care antigen rapid test (AG-rt) device in its performance compared to laboratory-based rt-PCR testing in COVID-19 suspected, symptomatic patients.

### Methods

For this prospective study, two specimens each of 541 symptomatic female (54.7%) and male (45.3%) patients aged between 18 and 95 years tested at five emergency departments (ED, n = 296) and four primary healthcare centres (PHC, n = 245), were compared, using AG-rt (positive/negative/invalid) and rt-PCR (positive/negative and cycle threshold, Ct) to diagnose SARS-CoV-2. Diagnostic accuracy, sensitivity, specificity, positive predictive values (PPV), negative predictive value (NPV), and likelihood ratios (LR+/-) of the AG-rt were assessed.

### Results

Differences between ED and PHC were detected regarding gender, age, symptoms, disease prevalence, and diagnostic performance. Overall, 174 (32.2%) were tested positive on

**Funding:** The trial was funded by Roche Austria to cover costs of the statistician and to provide material for the study.

**Competing interests:** The authors have read the journal's policy and have the following competing interests: Roche Diagnostics provided support for this study in the form of funds sent to the scientific association Science Center Donaustadt, which were used to cover the costs of the test materials and statistician. There are no patents, products in development or marketed products associated with this research to declare. This does not alter our adherence to PLOS ONE policies on sharing data and materials.

AG-rt and 213 (39.4%) on rt-PCR. AG correctly classified 91.7% of all rt-PCR positive cases with a sensitivity of 80.3%, specificity of 99.1%, PPV of 98.3, NPV of 88.6%, LR(+) of 87.8, and LR(-) of 0.20. The highest sensitivities and specificities of AG-rt were detected in PHC (sensitivity: 84.4%, specificity: 100.0%), when using Ct of 30 as cut-off (sensitivity: 92.5%, specificity: 97.8%), and when symptom onset was within the first three days (sensitivity: 82.9%, specificity: 99.6%).

## Conclusions

The highest sensitivity was detected with a high viral load. Our findings suggest that AG-rt are comparable to rt-PCR to diagnose SARS-CoV-2 in COVID-19 suspected symptomatic patients presenting both at emergency departments and primary health care centres.

## Introduction

Conventional diagnostic steps for infection with SARS-CoV-19 were epidemiological contact history, clinical impression, chest radiography, standard blood laboratory, and antigen detection by means of real-time polymerase chain reaction (rt-PCR). PCR remains the gold standard test for detection of SARS-CoV-2 infection [1].

As SARS-CoV-2 is being fought, testing not just of patients with suspected infection, but as well healthy individuals, takes places with rapid antigen test lateral flow devices [2, 3]. The way of handling is advantageous to PCR testing, as there is neither a need for laboratory staff nor for a laboratory environment, and the lateral flow device is rapid in application and timely superior to the PCR procedure.

It soon became apparent that in this pandemic the available capacities for PCR testing were by far from sufficient, and a feverish search for alternative and simpler detection methods began. Testing that takes a certain time can make up for significant additional efforts of organisation of patients in hospitals; and far beyond health systems, delay of testing effects societies as a whole. This regards nearly all spheres of life. Furthermore, the ideal test system is reliable, fast, easy to use, and affordable.

Besides PCR-testing, detection of antibodies against SARS-CoV-2 can play a role as well. Serology is generally available, however, serology seems only interpretable with the knowledge of patient's history and clinical appearance. IgA and IgM seem to quickly fade within 10 to 15 days, fade and thus, are not always be detectable, as opposed to IgG [4–6].

However, antigen tests tend to better detect SARS-CoV-2, the more virus load the nasopharyngeal mucus contains [7, 8]. In a meta-analysis by Dinnes et al. [9], five trials compared rt-PCR with 943 antigen tests were pooled. The average sensitivity was 56.2% (95% CI 29.5% to 79.8%), and the average specificity was 99.5% (95% CI 98.1% to 99.9%). More promising results were reported by Porte et al., who tested 82 rt-PCR-positive samples with another rapid antigen test and found a sensitivity of 93.9% [10].

The purpose of this study was to assess the performance of a novel CE-marked *in vitro* diagnostics (IVD) assay, the SARS-CoV-2 Rapid Antigen Test (Roche Diagnostics), for the detection of SARS-CoV-2 antigen. According to the manufacturer's manual [11], the antigen test shows 96.5% (95% CI 91.3% to 99.0%) of sensitivity and 99.7% (95% CI 98.2 to 99.9%) of specificity.

## Material and methods

### Patients

Patients were recruited consecutively between October 30, 2020 and December 13, 2020 at five emergency departments and four primary healthcare centres in Austria. The study was approved by four provincial ethics commissions (EK20-249-1020, GS1-EK-3/182-2020, 33–064 ex 20/21, ABT08GP-15681/2020-18). Signed informed consent was obtained from all participants.

591 symptomatic adults ($\geq$18 years) were included, who were willing to undergo sampling twice. Inclusion criteria were cough, fever, ageusia/anosmia, shortness of breath and sore throat. A total of 49 (8.3%) patients were excluded for the following reasons: asymptomatic (n = 12), children (<18years, n = 10), missing rt-PCR and/or AG-rt data (n = 8), unknown symptoms (n = 13), and symptom onset more than two weeks prior to testing (n = 6).

### Procedure

Two swabs per patient were taken by experienced medical staff. The first probe was analysed using the point-of-care device (SARS-CoV-2 Rapid Antigen Test (Roche Diagnostics). Outcome was recorded 15 minutes after sampling as positive, negative, or invalid. Only one case (1/542 = 0.2%), a 31 year old male patient with a sore throat two days prior testing and a negative rt-PCR, showed an invalid AG-rt reading, which was not included in analysis. All rt-PCR analyses using the second probe of each patient was conducted in hospital's laboratories or in other special laboratories. Rt-PCR results were collected as quantitative (Ct) and qualitative (positive or negative) parameters. Ct was reported in 202 of 213 cases.

### Statistical analysis

Descriptive statistics was used to describe the characteristics of patients. The distribution of the data was approximated by visual inspection of the histograms and the Kolmogorov Smirnov tests. Normally distributed data were calculated as mean value with standard deviation (SD), otherwise as median and range.

Continuous variables were compared between two groups with independent t-tests (parametric) or Mann-Whitney U-tests (non-parametric). Chi-square or Fisher's exact tests were applied to describe the relationship between proportions of categorical variables. Correlations between the continuous parameters were performed using Spearman's rho.

Percentage accuracy in classification, sensitivity, specificity, positive predictive value (PPV), and negative predictive value (NPV) were calculated. Positive (+) and negative (-) likelihood ratios (LR) were calculated using sensitivity and specificity. The larger LR(+), the greater the likelihood to be SARS-CoV-2 positive; and similarly, the smaller the LR(-), the lesser the likelihood to be SARS-CoV-2 positive. All values are presented with their 95% confidence interval (95% CI).

Statistical significance was set at a p-value of <0.05 (two-sided). All data were analysed with SPSS software (IMP Statistics Version 25; SPSS Inc, Chicago, IL) and MedCalc Statistical Software version 19.6.4 (MedCalc Software bv, Ostend, Belgium; https://www.medcalc.org; 2020).

## Results

Included in this prospective diagnostic study were 541 symptomatic patients of five ED (n = 296) and four PHC (n = 245), who were tested for SARS-CoV-2 using AG-rt and rt-PCR. The average age of the consecutively tested patients including 54.7% females and 45.3% males was 49.1±19.7years (range, 18-95years).

**Table 1. Comparison of demographic characteristics between emergency departments (ED) and primary healthcare centres (PHC).**

| Characteristic | ED (N = 296) | PHC (N = 245) | P Value |
|---|---|---|---|
| Male–n / total N (%) | 151 (51.0) | 88 (35.9) | <0.001* |
| Age—median years (min-max) | 58 (19–95) | 37 (18–77) | <0.001† |
| age groups–n (%) | | | |
| 18–29 years | 37 (12.5) | 66 (26.9) | <0.001* |
| 30–39 years | 28 (9.5) | 71 (29.0) | |
| 40–49 years | 36 (12.2) | 53 (21.6) | |
| 50–59 years | 53 (17.9) | 28 (11.4) | |
| 60–69 years | 43 (14.5) | 18 (7.3) | |
| 70–79 years | 58 (19.6) | 9 (3.7) | |
| 80–89 years | 36 (12.2) | 0 | |
| ≥90 years | 5 (1.7) | 0 | |
| Symptoms–median n (min-max) | 2 (1–5) | 1 (1–4) | <0.001† |
| fever–n (%) | 198 (66.9) | 65 (26.5) | <0.001* |
| cough–n (%) | 156 (52.7) | 108 (44.1) | 0.046* |
| sore throat–n (%) | 56 (18.9) | 150 (61.2) | <0.001* |
| dysgeusia/anosmia–n (%) | 42 (14.2) | 28 (11.4) | 0.341* |
| dyspnoea–n (%) | 125 (42.2) | 15 (6.1) | <0.001* |
| rhinitis–n (%) | 1 (0.3) | 0 | - |
| diarrhoea–n (%) | 2 (0.7) | 0 | - |
| others–n (%) | 7 (2.4) | 2 (0.8) | - |

Abbreviation: n and N, number; others including nausea, vomiting, fatigue, myalgia, and cephalea.

* Chi-square test;

† Mann-Whitney U-test;

‡ Fischer's exact test.

A comparison between demographic characteristics between patients tested at ED and PHC showed significant differences regarding gender, age, and some symptoms (Table 1). The main symptom for patients presenting in ED was fever, while more than 60% of patients at a PHC reported to have a sore throat as most common symptom.

Overall, 174 (32.2%) were tested positive on AG-rt and 213 (39.4%) on rt-PCR (Fig 1). AG correctly classified 91.7% [95%CI 89.0–93.9] of all rt-PCR positive cases with a sensitivity of 80.3% [95%CI 74.3–85.4], specificity of 99.1% [95%CI 97.4–99.8], PPV of 98.3 [95%CI 94.7–99.4], NPV of 88.6% [95%CI 85.5–91.0], LR(+) of 87.8 [28.4–271.3], and LR(-) 0.20 [0.15–0.26].

There were only three of 541 patients (0.6%) including two female aged 87 and 57 years and one 59-year old male with false positive AG-rt results. All three patients presented within five days of first symptoms' onset with fever and cough and one additionally with dyspnoea.

The false negative tested patients (7.8%; ED: 30 and PHC: 12) included 50% females with an average age of 53.0±17.2 years, the following symptoms: fever (n = 29); cough (n = 27); sore throat (n = 12); dyspnoea (n = 12); and dysgeusia/anosmia (n = 5) with a median symptom onset of 3 (range, 0–14 days), and an average Ct of 31.2±3.9.

Sensitivity and specificity were similar regarding gender (female: 81.1% and 99.0%; male: 79.4% and 99.3%) and symptoms (fever: 79.0% and 97.6%; cough: 78.2% and 97.9%; dyspnoea: 79.0% and 98.8%; sore throat: 80.0% and 100.0%). Prevalence of the disease (positive rt-PCR test) was 36.8% in female and 42.7% in male patients; in patients aged 18 to 45 years, 46 to 65

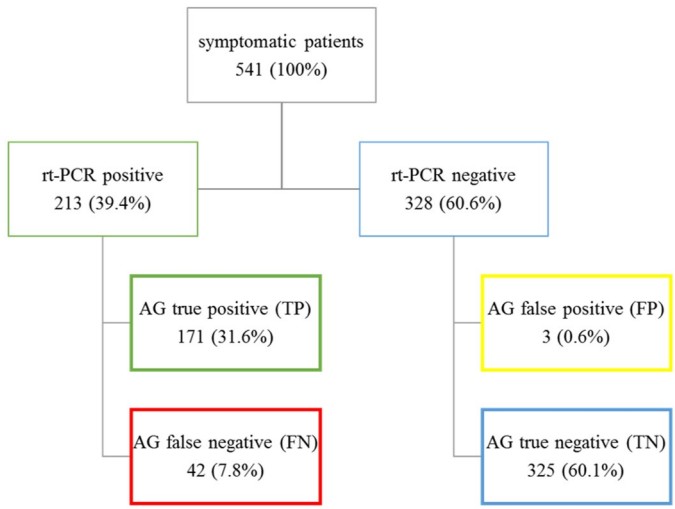

**Fig 1. Overall testing outcome.**

years, and older than 65 years, the prevalence was 33.9%, 45.7%, and 42.1%, respectively (S1 File).

Diagnostic performance of the AG-rt at ED and PHC are presented in Table 2. Interestingly, sensitivity was lower and there were more than twice false negatives in ED compared to PHC.

Sensitivities between ED and PHC regarding days of symptom onset are presented in Fig 2.

The majority of patients (72.6%) was tested within the first three days after symptom onset. Symptom onset within 3 days and between 4 and 7 days showed a sensitivity above 80%, while onset of symptoms between 8 and 14 days was associated with a far less sensitivity. However, days of onset did not correlate with Ct (Spearman's rho = 0.109; p = 0.124). Details regarding diagnostic performance of the AG-rt according symptom onset are presented in Table 3.

**Table 2. Diagnostic performance of the antigen rapid test (AG-rt) at emergency departments (ED) and primary healthcare centres (PHC).**

|  | ED (n = 296) | PHC (n = 245) |
|---|---|---|
| **True positive n (%)** | 106 (35.8) | 65 (26.5) |
| **False positive n (%)** | 3 (1.0) | 0 |
| **False negative n (%)** | 30 (10.1) | 12 (4.9) |
| **True negative n (%)** | 157 (53.1) | 168 (68.6) |
| **Disease prevalence (%) [95% CI]** | 46.0 [40.2–51.8] | 31.4 [25.7–37.7] |
| **Accuracy (%) [95% CI]** | 88.9 [84.7–92.2] | 95.1 [91.6–97.4] |
| **Sensitivity (%) [95% CI]** | 77.9 [70.0–84.6] | 84.4 [74.4–91.7] |
| **Specificity (%) [95% CI]** | 98.1 [94.6–99.6] | 100 [97.8–100.0] |
| **PPV (%) [95% CI]** | 97.3 [92.0–99.1] | 100 |
| **NPV (%) [95% CI]** | 84.0 [79.2–87.8] | 93.3 [89.3–95.9] |
| **LR(+) [95% CI]** | 41.6 [13.5–128.0] | - |
| **LR(-) [95% CI]** | 0.22 [0.16–0.31] | 0.16 [0.09–0.26] |

Abbreviation: CI, confidence interval; ED, emergency departments; LR, likelihood ratio; n, numbers; NPV, negative predictive value; PHC, primary health care centres; PPV, positive predictive value.

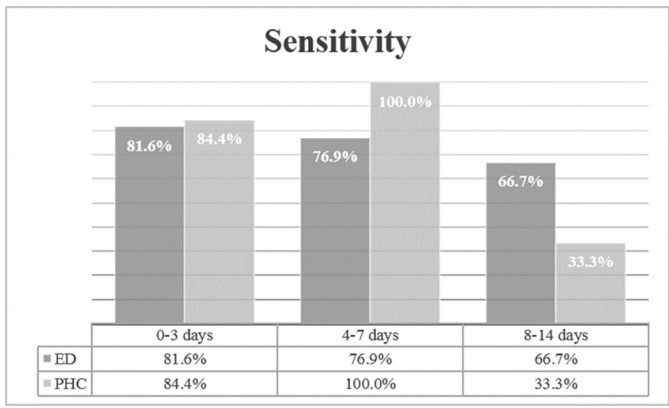

**Fig 2. Sensitivities (in % with 95% confidence interval in parenthesis) between ED and PHC regarding days of symptom onset.**

Sensitivities between ED and PHC regarding rt-PCR cut offs at 20, 25, 30, and 40 are presented in Fig 3.

Diagnostic performance of the AG-rt with rt-PCR cut-offs defined as positive (Ct = 1–39) and negative (Ct > 40), and at Ct values of 30, 25, and 20 are presented in Table 4. The highest sensitivity and specificity was detected when using Ct of 30 as cut-off.

Furthermore, significant lower Ct values were detected between TP (n = 165; 22.2±4.2) and FN (n = 37; 31.2±3.9; p<0.001; Fig 4).

## Discussion

Rt-PCR testing is the gold-standard procedure for SARS-CoV-2 infection. As its results are often not rapidly or timely available for every patient, the use of rt-PCR all too often is not

**Table 3. Diagnostic performance of the antigen rapid test (AG-rt) according symptom(s) onset.**

| AG-rt (n = 541) | Onset within 3 days (n = 393) | Onset 4–7 days (n = 98) | Onset 8–14 days (n = 50) |
|---|---|---|---|
| **True positive n (%)** | 116 (29.5) | 40 (40.8) | 15 (30.0) |
| **False positive n (%)** | 1 (0.3) | 2 (2.0) | 0 |
| **False negative n (%)** | 24 (6.1) | 9 (9.2) | 9 (18.0) |
| **True negative n (%)** | 252 (64.1) | 47 (48.0) | 26 (52.0) |
| **Disease prevalence (%) [95% CI]** | 35.6 [30.9–40.6] | 50.0 [39.7–60.3] | 48.0 [33.7–62.6] |
| **Accuracy (%) [95% CI]** | 93.6 [90.8–95.8] | 88.8 [80.8–94.3] | 82.0 [68.6–91.4] |
| **Sensitivity (%) [95% CI]** | 82.9 [75.6–88.7] | 81.6 [68.0–91.2] | 62.5 [40.6–81.2] |
| **Specificity (%) [95% CI]** | 99.6 [97.8–100.0] | 95.9 [86.0–99.5] | 100.0 [86.8–100.0] |
| **PPV (%) [95% CI]** | 99.2 [94.3–99.9] | 95.2 [83.6–98.7] | 100 |
| **NPV (%) [95% CI]** | 91.3 [87.9–93.8] | 83.9 [74.3–90.4] | 74.3 [63.3–82.9] |
| **LR(+) [95% CI]** | 209.6 [29.6–1484.6] | 20.0 [5.1–78.2] | - |
| **LR(-) [95% CI]** | 0.17 [0.12–0.25] | 0.19 [0.11–0.35] | 0.38 [0.22–0.63] |

Abbreviation: AG-rt, antigen rapid test; CI, confidence interval; LR, likelihood ratio; n, numbers; NPV, negative predictive value; PPV, positive predictive value.

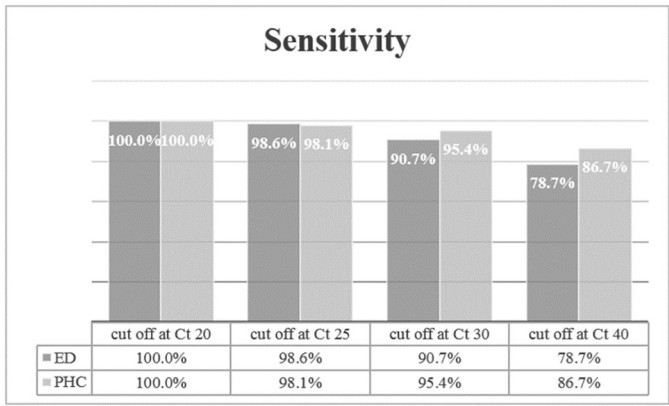

**Fig 3. Sensitivities (in % with 95% confidence interval in parenthesis) between ED and PHC regarding rt-PCR cut offs.**

adequate neither in emergency departments nor in general practitioner's settings. Patient's flows are crucial in the management of the disease, and so is the diagnostic flow in medical institutions such as emergency departments or primary health care centres.

We found the SARS-CoV-2 Rapid Antigen Test (Roche Diagnostics) to produce an overall sensitivity of 80.3% and specificity of 99.1% compared with rt-PCR, both in emergency departments and primary health care centres. From symptom onset days 0 to 7, the sensitivity was much better with 82.2%, whereas it reached 62.5% with disease onset from days 8 to 14. Sensitivities were higher with lower PCR cycle threshold numbers.

Our results differ from the numbers claimed by the manufacturer, who reported a sensitivity of 96.5% and a specificity of 99.7%, who might have used specimens displaying higher viral loads [11].

**Table 4. Diagnostic performance of the antigen rapid test (AG-rt) according to various cut-offs.**

| AG-rt (n = 532*) | rt-PCR | rt-PCR Ct30 | rt-PCR Ct25 | rt-PCR Ct20 |
|---|---|---|---|---|
| **Cut-off** | | | | |
| positive | Ct = 1–39 | Ct ≤ 30 | Ct ≤ 25 | Ct ≤ 20 |
| negative | Ct > 40 | Ct > 30 | Ct > 25 | Ct > 20 |
| **True positive n (%)** | 165 (31.0) | 160 (30.1) | 123 (23.1) | 51 (9.6) |
| **False positive n (%)** | 3 (0.6) | 8 (1.5) | 45 (8.5) | 117 (22.0) |
| **False negative n (%)** | 37 (7.0) | 13 (2.4) | 2 (0.4) | 0 |
| **True negative n (%)** | 327 (61.4) | 351 (66.0) | 362 (68.0) | 364 (68.4) |
| **Disease prevalence (%) [95% CI]** | 38.0 [33.8–42.3] | 32.5 [28.6–36.7] | 23.5 [20.0–27.3] | 9.6 [7.2–12.4] |
| **Accuracy (%) [95% CI]** | 92.5 [89.9–94.6] | 96.1 [94.0–97.5] | 91.2 [88.4–93.4] | 78.0 [74.2–81.5] |
| **Sensitivity (%) [95% CI]** | 81.7 [75.7–86.8] | 92.5 [87.5–95.9] | 98.4 [94.3–99.8] | 100 [93.0–100.0] |
| **Specificity (%) [95% CI]** | 99.1 [97.4–99.8] | 97.8 [95.7–99.0] | 88.9 [85.5–91.8] | 75.7 [71.6–79.5] |
| **PPV (%) [95% CI]** | 98.2 [94.7–99.4] | 95.2 [91.0–97.6] | 73.2 [67.5–78.3] | 30.4 [27.1–33.8] |
| **NPV (%) [95% CI]** | 89.8 [86.9–92.2] | 96.4 [94.1–97.9] | 99.5 [97.9–99.9] | 100 |
| **LR(+) [95% CI]** | 89.9 [29.1–277.7] | 41.5 [20.9–82.5] | 8.9 [6.8–11.7] | 4.1 [3.5–4.8] |
| **LR(-) [95% CI]** | 0.18 [0.14–0.25] | 0.08 [0.05–0.13] | 0.02 [0.00–0.07] | 0 |

Abbreviation: AG-rt, antigen rapid test; CI, confidence interval; Ct, cycle threshold; LR, likelihood ratio; n, numbers; NPV, negative predictive value; PPV, positive predictive value; rt-PCR, reverse transcription polymerase chain reaction.

* No Ct was available in 11 samples.

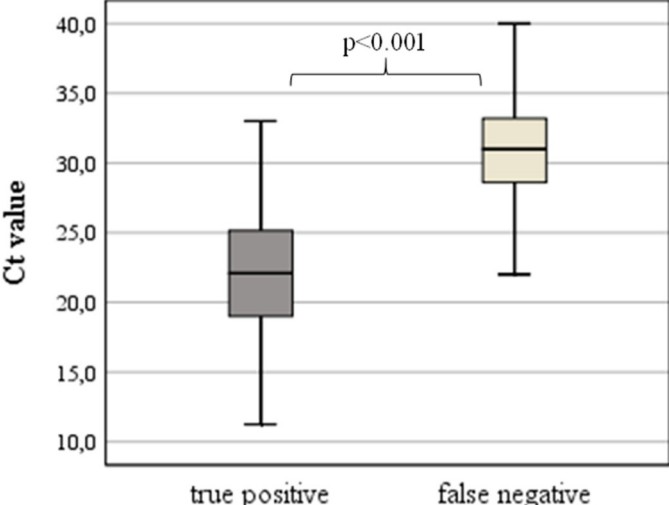

**Fig 4. Boxplot showing cycle threshold (Ct) values of true positive (n = 165) and false negative (n = 37).**

Sensitivities of antigen assays in previously published trials including a meta-analysis showed a wide range from 45% to 86% [9, 10, 12–14]. Direct comparison exposes varieties in test systems, onset of disease, performance of the procedure, presence of symptoms, testing institutions, and others. Notably, in our trial results including sensitivities and specificities differed between ED and PHC. We did not find a conclusive explanation for this fact; staffs and their respective training did not differ in any of the centres substantially.

Only in few patients (0.6%) a false positive result was detected with the IVD compared to PCR testing. The implication of this number however, is that those patients obviously apparently are to be sequestered into quarantine jointly with patients with true positive results, as long as PCR test results are pending. This requires a careful epidemiological reflection, when mass testing is performed.

The rate of false negative patients remained under 10%. None of these patients (n = 37) had a Ct under 22; in 4 patients a Ct of 23 to 25, and in a further 8 patients a Ct of 27 to 30 was detected; all other patients (n = 25) had a Ct above 30. This underlines the correlation of a virus detection by means of the device and the viral load.

One of the limitations is the sole inclusion of symptomatic cases and not asymptomatic persons. Actually, the purpose of the device under investigation is indeed the testing of symptomatic persons with a suspected SARS-CoV-2 infection, which corresponds to the approval of the device. No severity of symptoms and progress was considered for our trial.

Additionally, despite being considered as the gold standard, PCR testing is not 100% accurate and test quality crucially depends on the quality of manual sampling of specimen [15].

## Conclusion

This prospective study demonstrated a performance of the SARS-CoV-2 Rapid Antigen Test (Roche Diagnostics) with an overall sensitivity of 80.3% compared to rt-PCR, which, in case of a negative result, needs to be interpreted together with the duration of the disease at the time of testing, the viral load, and likely the diligence of the generation of the specimen.

## Supporting information

**S1 File. Diagnostic performance of the antigen rapid test (AG-rt) of females, males, age groups: Young (18-45y), middle (46-65y), old (66y plus), and symptoms: Fever, cough, dyspnoea, sore throat.**
(DOCX)

## Author Contributions

**Conceptualization:** Rainer Thell, Verena Kallab, Christian Sebesta, Stefan Winkler, Marton Szell.

**Data curation:** Rainer Thell, Verena Kallab, Wolfgang Weinhappel, Wolfgang Mueckstein, Lukas Heschl, Martina Heschl, Stefan Korsatko, Franz Toedling, Amelie Blaschke, Theresa Herzog, Anna Klicpera, Clara Koeller, Moritz Haugk, Anna Kreil, Alexander Spiel, Philipp Kreuzer, Robert Krause, Marton Szell.

**Formal analysis:** Rainer Thell, Theresa Herzog, Christian Sebesta, Brenda Laky.

**Funding acquisition:** Christian Sebesta.

**Investigation:** Rainer Thell, Brenda Laky, Marton Szell.

**Methodology:** Rainer Thell, Christian Sebesta, Stefan Winkler, Brenda Laky, Marton Szell.

**Project administration:** Brenda Laky.

**Resources:** Marton Szell.

**Software:** Brenda Laky.

**Supervision:** Stefan Winkler, Marton Szell.

**Visualization:** Rainer Thell.

**Writing – original draft:** Rainer Thell.

**Writing – review & editing:** Rainer Thell, Brenda Laky, Marton Szell.

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
