## [Decision Letter · Decision Letter 0]

9 Jun 2021

PONE-D-21-15345

Evaluation of a novel, rapid antigen detection test for the diagnosis of SARS-CoV-2

PLOS ONE

Dear Dr. Thell,

Thank you for submitting your manuscript to PLOS ONE. After careful consideration, we feel that it has merit but does not fully meet PLOS ONE’s publication criteria as it currently stands. Therefore, we invite you to submit a revised version of the manuscript that addresses the points raised during the review process.

We look forward to receiving your revised manuscript.

Kind regards,

Etsuro Ito

Academic Editor

PLOS ONE

Journal Requirements:

3. Please include the full name of the ethics committee(s) that approved your study in the manuscript Methods.

5. Please upload a copy of Supplement Table S1 which you refer to in your text on page 9.

6.Thank you for stating the following in the Funding Section of your manuscript:

"Roche diagnostics provided the SARS-CoV-2 Rapid Antigen Test (Roche Diagnostics).

283 All authors declare that they have no conflict of interest. Our group received funding for this

284 research project from Roche Diagnostics."

"no"

Additionally, because some of your funding information pertains to [commercial funding//patents], we ask you to provide an updated Competing Interests statement, declaring all sources of commercial funding.

In your Competing Interests statement, please confirm that your commercial funding does not alter your adherence to PLOS ONE Editorial policies and criteria by including the following statement: "This does not alter our adherence to PLOS ONE policies on sharing data and materials.” as detailed online in our guide for authors  http://journals.plos.org/plosone/s/competing-interests.  If this statement is not true and your adherence to PLOS policies on sharing data and materials is altered, please explain how.

Please include the updated Competing Interests Statement and Funding Statement in your cover letter. We will change the online submission form on your behalf.

Reviewers' comments:

Reviewer's Responses to Questions

**Comments to the Author**

1. Is the manuscript technically sound, and do the data support the conclusions?

Reviewer #1: Yes

Reviewer #2: Partly

2. Has the statistical analysis been performed appropriately and rigorously? 

Reviewer #1: I Don't Know

Reviewer #2: N/A

3. Have the authors made all data underlying the findings in their manuscript fully available?

Reviewer #1: Yes

Reviewer #2: No

4. Is the manuscript presented in an intelligible fashion and written in standard English?

Reviewer #1: Yes

Reviewer #2: Yes

5. Review Comments to the Author

Reviewer #1: This is part of an important body of literature, and should be published by PLOS. However, it should be noted that (like essentially every publication in this field of this type) this paper does not strongly meet the public health need, which boils down to one question: Should we use this test?

The conclusions are in the Results section (which is fine), but the statements in the Conclusion section that the two tests are "comparable" is vacant (obviously, a test that has a PPV of zero can be "compared" to a test with a PPV of 100). Likewise, the statement that "the highest sensitivity was detected with a high viral load" is, of course, unexceptional. The key point of this paper, which is VERY important and MUST be published, likely to be highlighted, is buried in the discussion: "Our results differ from the numbers claimed by the manufacturer, who reported a sensitivity of

251 96.5% and a specificity of 99.7%". We would prefer something more direct, like: "The results reported here suggest poorer performance than claimed by the manufacturer".

The public health questions begin with the recognition that infected persons who might enter a public space have a range of viral loads in their "expectorant", the material that they might broadcast while speaking, coughing, sneezing, exhaling. The risk that they present to the public space is (likely) an increasing function of that viral load. That risk may have a cutoff, that is, a viral load that is high enough to remain detectable by a specific test, but sufficiently low that expectorant does not present a risk. The question that everyone asks about a test being discussed is: Does the test meet the cutoff? Or is the sensitivity of the test so poor that it leaves undetected a fraction of the persons who present a risk.

Of course, COVID-19 is especially problematic in its asymptomatic carriers. This paper, of course, looks at only symptomatic people. The manufacturer perhaps looked at even MORE symptomatic people. Thus, the "statistical analysis" beloved by biostatisticians (and PLOS reviewers/editors) is overwhelmed by a factor not captured, or captur-able, by any statistical analysis. A systematic sampling bias.

Here, this review runs counter to the "culture" in this field. That culture looks for the "statistical analysis". A paper is accepted in the field if it applies correctly a few "tests", reports chi squares, and so on, all beloved by statisticians. As a result we get lots of papers that would get an A grade in a statistics course, but provide little guidance to PHS officials.

Now, I am recommending publication for two reasons:

1. What we PHS people want, we cannot get. Regulatory agencies, IRBs, and the entire "business as usual" government-medical bureaucracy of "experts" place obstacles to the needed studies. Specifically, such studies need to include asymptomatic people selected randomly and by surprise. These are people who are NOT enrolled in studies where filling out paperwork is taken to be the equivalent of "ethics". However, there is no reason to reject this paper and its important conclusion because "experts" are preventing (and have done so for more than a year now) the studies that are needed to manage this pandemic.

2. We do not have an understanding of the function that relates forward transmission risk to viral load. Thus, we do not know the cutoff. This means that even if the "experts" were to allow a truly random sample unfiltered by an IRB-style enrollment program, we could not compare the cutoff (if one exists) with the sensitivity of the test.

Reviewer #2: The study is evidently, a precise and concise research. It was fascinating as the authors proposed a fast and feasible approach (SARS-CoV-2 Rapid Antigen Test) the SARS-CoV-2 infection can be detected, compared to rt-PCR. The methods that were adopted for the study, including the omission cases where certain criteria were not met by any study participant, was apt and impressive.

However, I have minor concerns that I would love the authors to address for clarifications.

The age group was between 18-90, as the selected population cohort for the study, Is there a possibility this detection method could work for those outside that age group especially children and teenagers. Kindly, give your reasons and if there are future research that will be conducted on that.

For the statistical analysis, I would advise the authors to provide a convincing graphical displays in colours to easily different different scenario. This also applies to the flow chart provided by the authors.

Lastly, I observed few grammatical errors and punctutaions.

Asides these, i think it is a good and a novel approach that the authors have adopted.

6. PLOS authors have the option to publish the peer review history of their article (what does this mean?). If published, this will include your full peer review and any attached files.

Reviewer #1: No

Reviewer #2: **Yes: **Onyeka S. Chukwudozie

---

## [Editor Report · Decision Letter 1]

21 Oct 2021

Evaluation of a novel, rapid antigen detection test for the diagnosis of SARS-CoV-2

PONE-D-21-15345R1

Dear Dr. Thell,

We’re pleased to inform you that your manuscript has been judged scientifically suitable for publication and will be formally accepted for publication once it meets all outstanding technical requirements.

Kind regards,

Etsuro Ito

Academic Editor

PLOS ONE

---

## [Editor Report · Acceptance letter]

17 Nov 2021

PONE-D-21-15345R1 

Evaluation of a novel, rapid antigen detection test for the diagnosis of SARS-CoV-2 

Dear Dr. Thell:

I'm pleased to inform you that your manuscript has been deemed suitable for publication in PLOS ONE. Congratulations! Your manuscript is now with our production department. 

Kind regards, 

on behalf of

Prof. Etsuro Ito 

Academic Editor

PLOS ONE